# Comparative analysis of web-based programs for single amino acid substitutions in proteins

**Arunabh Choudhury[1], Taj Mohammad[2], Farah Anjum[3], Alaa Shafie[3], Indrakant K. Singh[4], Bekhzod Abdullaev[5], Visweswara Rao Pasupuleti[6,7,8], Mohd Adnan[9], Dharmendra Kumar Yadav[10]\*, Md. Imtaiyaz Hassan[2]\***

**1** Department of Computer Science, Jamia Millia Islamia, Jamia Nagar, New Delhi, INDIA, **2** Centre for Interdisciplinary Research in Basic Sciences, Jamia Millia Islamia, Jamia Nagar, New Delhi, INDIA, **3** Department of Clinical Laboratory Sciences, College of Applied Medical Sciences, Taif University, Taif, Saudi Arabia, **4** Molecular Biology Research Lab, Department of Zoology, Deshbandhu College, University of Delhi, Kalkaji, New Delhi, India, **5** Scientific Department, Akfa University, Tashkent, Uzbekistan, **6** Department of Biomedical Sciences and Therapeutics, Faculty of Medicine & Health Sciences, Universiti Malaysia Sabah, Kota Kinabalu, Sabah, Malaysia, **7** Department of Biochemistry, Faculty of Medicine and Health Sciences, Abdurrab University, Pek-anbaru, Riau, Indonesia, **8** Centre for International Collaboration and Research, Reva University, Rukmini Knowledge Park, Kattigenahalli, Yelahanka, Bangalore, Karnataka, India, **9** Department of Biology, College of Science, University of Hail, Hail, Saudi Arabia, **10** College of Pharmacy, Gachon University of Medicine and Science, Yeonsu-gu, Incheon City, Korea

\* dharmendra30oct@gmail.com (DKY); mihassan@jmi.ac.in (MIH)

**Data Availability Statement:** All relevant data are within the manuscript and its Supporting Information files.

## Abstract

Single amino-acid substitution in a protein affects its structure and function. These changes are the primary reasons for the advent of many complex diseases. Analyzing single point mutations in a protein is crucial to see their impact and to understand the disease mechanism. This has given many biophysical resources, including databases and web-based tools to explore the effects of mutations on the structure and function of human proteins. For a given mutation, each tool provides a score-based outcomes which indicate deleterious probability. In recent years, developments in existing programs and the introduction of new prediction algorithms have transformed the state-of-the-art protein mutation analysis. In this study, we have performed a systematic study of the most commonly used mutational analysis programs (10 sequence-based and 5 structure-based) to compare their prediction efficiency. We have carried out extensive mutational analyses using these tools for previously known pathogenic single point mutations of five different proteins. These analyses suggested that sequence-based tools, PolyPhen2, PROVEAN, and PMut, and structure-based web tool, mCSM have a better prediction accuracy. This study indicates that the employment of more than one program based on different approaches should significantly improve the prediction power of the available methods.

## Introduction

Non-synonymous single nucleotide polymorphism (nsSNP) in the genome introduces a single amino acid change in the protein sequence, which may or may not affect a protein in terms of

**Funding:** This work was supported by Taif University Researchers Supporting Project Number (TURSP-2020/131), Taif University, Taif, Saudi Arabia.

**Competing interests:** The authors have declared that no competing interests exist.

structure and subsequent function. An amino acid substitution on a protein can have several effects, including loss or gain in function, alteration of the catalytic site, structural instability, protein aggregation, or abnormal folding [1]. Also, missense mutations can impact the pre-translational and post-translational processes. Many human genetic disorders arise because of amino acid substitutions [2]. Over the past few decades, a considerable emphasis was given to analyzing single-point protein mutations to determine the effects and to understand the molecular mechanism [3–7]. This has produces many resources, including several databases and web-based tools that mostly focus on human mutations [8].

Many databases have been created to store the information about mutations of human and other organisms' genomes which serve as a starting point of mutation analysis. Most of the SNP data is deposited in The Single Nucleotide Polymorphism Database (dbSNP, http://www.ncbi.nlm.nih.gov/SNP/) [9], and it serves as the primary source for retrieval of single nucleotide polymorphisms. Ensemble (https://www.ensembl.org/) [10] is another large database that stores information about human and other organisms' genetic variations, and it also gives information about the pathogenesis of the variations. Other databases include Human Gene Mutation Database (HGMD, http://www.hgmd.cf.ac.uk/ac/index.php) [11], ClinVar (https://www.ncbi.nlm.nih.gov/clinvar/) [12], Online Mendelian Inheritance in Man (OMIM, http://www.ncbi.nlm.nih.gov/sites/entrez?db=omim) [13], the Pharmacogenetics Knowledge Base (PharmGKB, http://www.pharmgkb.org/) [14], etc.

Many bioinformatics tools have been developed to analyze the impact of missense mutations. Different approaches have been applied to the development of the tools (**Fig 1**). The two broad categories for mutation analysis are the sequence-based and structure-based approaches. Both approaches use several factors that affect the protein structure and function. The sequence-based study uses various analyzing methods, including cellular localization, aggregation, disorder, functional effects, and stability [1]. The structure-based approach is mainly based on the free energy calculation. It considers electrostatic changes, steric effects, interresidue contacts, disorder, functional effects, and stability. Sequence conservation is another vital component of SNP analysis as disease-causing mutations frequently occur in evolutionarily conserved regions [15–17]. Substitution of an amino acid increases the probability of protein

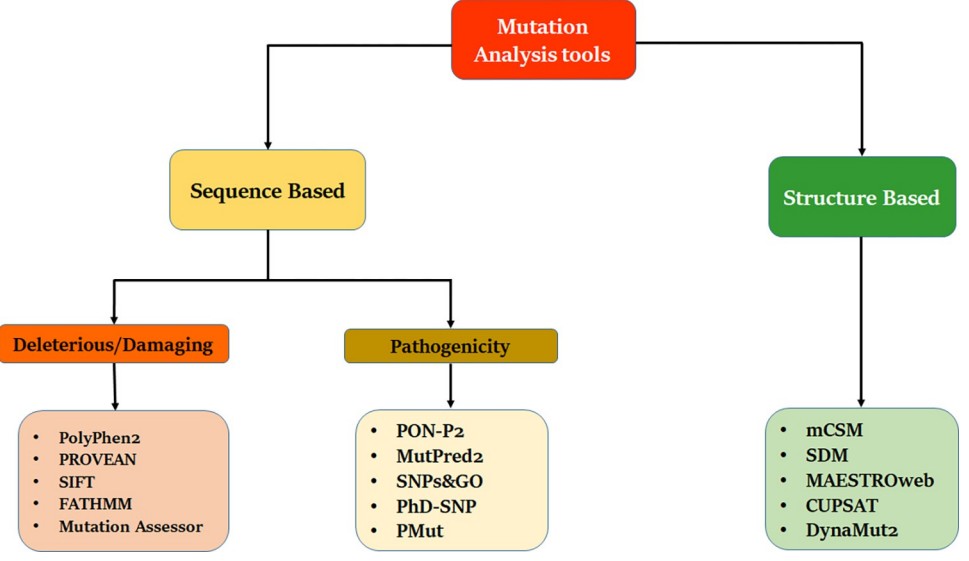

**Fig 1. Graphical representation of tools used in this study for comparison.**

getting aggregated which is involved in several neurodegenerative diseases [18, 19]. Amino acid substitution can introduce disorder in the protein structure. It can be estimated using amino acid composition, energy profiles and physicochemical properties, specific sequence patterns, missing X-ray coordinates, and B-factors. Changes in the electrostatic potential due to a substitution can affect the ligand-binding ability and folding mechanism. Phylogenetic information is also a key component in the prediction process.

The tools use different scoring matrices (such as BLOSSUM62) [20] to calculate whether a mutation has a functional or structural impact on a protein or not [17]. For a given mutation, each tool provides a score to indicate the damaging probability. In our study, we have performed a comparative analysis of 15 different web tools, out of which 10 are sequence-based, and five are structure-based. To compare the tools, we have taken previously confirmed disease-causing mutations and some functionally impactful/damaging mutations of five proteins and analyzed them through all fifteen tools (Table 1).

## Materials and methods

We have carried out an extensive mutational analysis of single point mutations of five different proteins through the 15 different sequence and structure-based tools. The proteins are Parkinson's disease protein 7 (PARK7), E3 ubiquitin-protein ligase parkin (PARK2), Presenilin-1 (PESN1), GTPase HRas (HRAS), and Runt-related transcription factor 1 (RUNX1). We have taken only those mutations which were already found to affect the protein function and structure. Sequences and the non-synonymous mutations of these proteins are collected from the UniProt database [21]. All the mutations are associated with a disease or have an altering effect on the protein, i.e., all the mutations are damaging to the protein. The structure of each protein was downloaded from the RCSB Protein Data Bank (PDB) [22]. For PARK7, PARK2, PESN1, HRAS and RUNX1, we have analyzed 18, 38, 199, 32 and 51 single point mutations, respectively. The UniProt and PDB IDs of the five proteins are given in Table 2.

We reviewed fifteen different tools, of which 10 are sequence-based and 5 are structure-based. We have also performed single-point mutation analysis to estimate their performance. PolyPhen2, PROVEAN, FATHMM, SIFT, Mutation Assessor, PON-P2, SNPs & GO,

**Table 1. Tools for the analysis of single amino acid substitutions.**

| Tool | URL | Prediction | Reference |
|---|---|---|---|
| PolyPhen2 | http://genetics.bwh.harvard.edu/pph2/ | Damaging or benign | [39] |
| PROVEAN | http://provean.jcvi.org/ | Deleterious or neutral | [24] |
| SIFT | http://sift.jcvi.org/ | Damaging or tolerated | [26] |
| FATHMM | http://hathmm.biocompute.org.uk | Damaging or tolerated | [40] |
| Mutation Assessor | http://mutationassessor.org/r3/ | Functionally impactful or neutral | [27] |
| PON-P2 | http://structure.bmc.lu.se/PON-P2/ | Pathogenicity prediction | [28] |
| MutPred2 | http://mutpred.mutdb.org | Pathogenicity prediction | [41] |
| SNPs & GO | https://snps-and-go.biocomp.unibo.it | Disease-causing or neutral | [42] |
| PhD-SNP | https://snps.biofold.org/phd-snp/phd-snp.html | Disease-causing or neutral | [43] |
| PMut | http://mmb.irbbarcelona.org/PMut | Disease-causing or neutral | [44] |
| mCSM | http://biosig.unimelb.edu.au/mcsm/ | Stability prediction | [45] |
| SDM | http://marid.bioc.cam.ac.uk/sdm2 | Stability prediction | [34] |
| MAESTROweb | https://pbwww.che.sbg.ac.at/maestro/web | Stability prediction | [46] |
| CUPSAT | http://cupsat.tu-bs.de/index.jsp | Stability prediction | [36] |
| DynaMut2 | http://biosig.unimelb.edu.au/dynamut2/ | Stability prediction | [37] |

Table 2. UniProt IDs and PDB IDs of each protein.

| Protein | UniProt ID | PDB ID |
|---|---|---|
| Parkinson's disease protein 7 (PARK7) | Q99497 | 1P5F |
| E3 ubiquitin-protein ligase parkin (PARK2) | O60260 | 5C1Z |
| Presenilin-1 (PESN1) | P49768 | 6IYC |
| GTPase HRas (HRAS) | P01112 | 4Q21 |
| Runt-related transcription factor 1 (RUNX1) | Q01196 | 1E50 |

PhD-SNP, MutPred2 and PMut are sequence-based and mCSM, SDM, MAESTROweb, CUP-SAT and DynaMut2 are structure-based tools.

## PolyPhen2

Polymorphism phenotyping (PolyPhen-2) is a sequence-based tool. The FASTA file is given as an input for the protein sequence [23]. To calculate the damaging probability of a mutation, it compares the physical properties of the wild-type and mutant variant. It incorporates multiple sequence alignment, and a machine learning-based classifier developed for high throughput NGS data analysis. PolyPhen2 derives Position-Specific Independent Count (PSIC) score for the variant and then estimates the difference of PSIC between mutant and the wild-type. For a PSIC score greater than 0.09, the tool considers a mutation to be deleterious.

## PROVEAN

The protein variation effect analyzer (PROVEAN) calculates the functional consequence of a single amino acid substitution on the protein [24]. PROVEAN categorizes mutations as deleterious or neutral; a mutation with a PROVEAN score of <-2.5 is deleterious, whereas mutations with scores >-2.5 are considered neutral. PROVEAN web server comprises three tools, PROVEAN Protein (includes any species), PROVEAN Protein Batch and PROVEAN Genome Variants (specifically for mouse and human). The PROVEAN Protein Batch tool also returns the result of SIFT tool, and it can process a large number of protein variants. The input for this program takes amino acid substitution and supports public domain protein identifiers from NCBI RefSeq, UniProt, and Ensembl.

## FATHMM

Functional Analysis through Hidden Markov Model (FATHMM) is a web-based tool for predicting the functional consequences of coding and non-coding variants in the human genome [25]. The coding variants can be analyzed for inherited diseases, cancer and specific diseases. FATHMM is comprised of two algorithms: unweighted and weighted. The unweighted method is based on sequence conservation, and the weighted method is a combination of sequence conservation and pathogenicity weights. The unweighted method searches conserved residues through the amino acid probabilities of various Hidden Markov Models (HMMs) representing the alignment of protein domains that are conserved and homologous sequences. The weighted method assigns pathogenicity weights that correlate with disease-causing amino acid substitutions, with sequence conservation found through searching HMMs.

## SIFT

Sorting Intolerant from Tolerant (SIFT) is a web-server that determines whether single amino acid substitutions on a protein are deleterious or not. The tool considers sequence similarity

and physical properties of the amino acid to calculate the damaging probability. A SIFT score of less than or equal to 0.05 indicates an intolerable mutation [26].

## Mutation assessor

Mutation Assessor is a sequence-based tool to predict the functional consequences of amino-acid substitutions in proteins. The Mutation Assessor depends upon multiple sequence alignment and amino acid residues that are evolutionarily conserved. The input of this tool includes UniProt protein accession or NCBI Refseq protein ID. It categorizes the protein variants as high, medium, low or neutral for damaging impacts. It returns the FI score for each variant. A variant with an FI score greater than 2.00 is predicted as a deleterious variant [27].

## PON-P2

PON-P2 is another web-based classifier for protein variants, and it uses a machine-learning-based approach. This tool differentiates the amino acid substitutions into pathogenic, neutral and unknown classes. It is a fast tool as it analyzes a large amount of variant data in less time in a highly efficient manner. This tool considers evolutionary sequence conservation, biochemical attributes and physical attributes of a protein. It also uses functional annotations and Gene Ontology (GO) annotations based on availability. The input of PON-P2 needs amino acid substitutions and one of Ensembl, Entrez or UniProtKB/Swiss-Prot accession ID [28].

## MutPred2

MutPred2 is a web-server that categorizes a mutation as disease-associated or neutral [29]. It estimates the molecular mechanism of pathogenicity of an amino acid substitution using a machine-learning-based technique. This tool considers fifty different protein properties to calculate the effect of the substitutions. For a pathogenic mutation, the MutPred2 score is greater than 0.5.

## SNPs & GO

SNPs & GO is a support vector machine (SVM) based web-server to identify deleterious single amino acid substitutions [30]. The SVM-based classifier consists of a single SVM that takes input protein sequence, profile and functional information. It uses GO annotations to classify a missense variant into disease-related or neutral. It requires amino acid sequence/SwissProt code, GO terms and amino acid substitutions as input. An SNPs & GO score of more than 0.5 indicates a disease-causing mutation. This tool also gives the result of PANTHER and PhD-SNP.

## PhD-SNP

Predictor of human Deleterious Single Nucleotide Polymorphisms (PhD-SNP) also uses SVM based classifier to classify the disease-associated variants [31]. Sequence and profile information is used in the classification process of the amino acid substitutions into neutral and disease-associated. The sequence profile is calculated using an input vector derived from wild-type (WT) and mutant amino acid frequencies, the number of aligned sequences, and the conservation score in the substituted location. A PhD-SNP score of more than 0.5 indicates a disease-causing mutation.

## PMut

Mutations that are associated with disease phenotype are identified using the PMut web server. A neural network-based method is used to train the classifier of PMut, and it uses the manually curated protein sequence data from the SwissProt database. Sequence conservation and physiochemical attributes of amino acids are used as the main features. People can also generate predictors for some protein families in the new version of the tool, and previously predicted results are also deposited in the webserver. If the PMut score for an amino acid substitution is greater than 0.05, then the variant is pathogenic [32].

## mCSM

mCSM predicts the stability of an amino acid substitution using a graph-based approach. The prediction method is trained with the environment derived from the atomic distance patterns of all the amino acid residues. It can estimate destabilizing probabilities for various protein structures and understand disease-associated variants. For a mutation that destabilizes a protein structure, the mCSM score ($\Delta\Delta G$) is less than 0 [33].

## SDM

Site-Directed Mutator (SDM) evaluates the protein stability upon single point mutations. Environment-specific amino acid substitution tables with parameters like packing density and residue length and PDB coordinate files are used to determine the stability of a mutant protein. SDM was tested with 2690 amino acid substitution from 132 different 3D structures of proteins. For a destabilizing amino acid substitution, the predicted $\Delta\Delta G$ is greater than 0 [34].

## MAESTROweb

MAESTRO is a multi-agent stability prediction web tool that calculates the free energy change on protein unfolding. The free energy difference ($\Delta\Delta G$) between the WT and mutant protein is calculated to determine the stability upon change in amino acid residues. The tool can evaluate both predetermined and modeled PDB coordinate files, although prediction accuracy for modeled structures are less efficient. For a mutation that has a destabilizing impact on a protein structure, the MAESTRO score is less than zero [35].

## CUPSAT

Cologne University Protein Stability Analysis Tool (CUPSAT) is a web-server to estimate changes in protein stability upon mutation [36]. The tool consists of a prediction model based on torsion angle distribution and potentials of the amino acid atoms. It assesses the amino acid environment around the substituted position. Secondary structure specificity and solvent accessibility are also used to determine the amino acid environment. In CUPSAT, the amino acid atom potentials of 40 amino acid atoms from Melo-Feytmans are used to construct the radial pair distribution function. CUPSAT gives the stability prediction upon mutation for all the amino acid mutations for a specific position. It can also predict custom PDB structures.

## DynaMut2

DynaMut2 is a protein stability prediction tool that combines Normal Mode Analysis (NMA) techniques to capture protein motion and graph-based signatures to represent the WT environment [37]. The data for the amino acid substitutions were taken from ProTherm. For stability prediction upon single point mutation, each mutation was modeled using many properties, including WT residue environment, protein dynamics (NMA), substitution

propensities and contact potential scores, interatomic interactions and graph-based signatures method. These methods were then used to train the machine learning algorithm. DynaMut2 can give predictions for single and multiple mutations. We have used the single mutation prediction feature for our analysis.

## Results and discussion

We have performed mutation analysis of all the five proteins through the 15 sequence and structure-based tools to estimate their performance (**S1 Data**). For PolyPhen2, we have used the batch query feature. In the batch query, several mutations can be predicted at once. Poly-Phen2 categorizes the mutations into damaging and benign classes. The predicted damaging mutations for PARK7, PARK2, PESN1, HRAS and RUNX1 are 66.67%, 92.11%, 96.09%, 78.13% and 100%, respectively. PolyPhen2 predicted an average of 86.60% damaging mutation from the five proteins. The predicted damaging mutations by PROVEAN for PARK7, PARK2, PESN1, HRAS and RUNX1 were 72.22%, 71.05%, 88.83%, 100%, and 100% respectively. The predicted damaging mutations by SIFT for PARK7, PARK2, PESN1, HRAS and RUNX1 were 66.67%, 78.95%, 90.50%, 90.63%, and 100%, respectively. PROVEAN predicted an average of 86.42% variants as damaging mutations, and SIFT predicted an average of 85.35% substitutions as damaging mutations from the five proteins (**Table 3**). For FATHMM analysis, we used the inherited disease feature under coding variants applying an unweighted algorithm. FATHMM gave 64.39% average damaging mutation for the five proteins. Mutation Assessor analyses predicted an average of 76.46% mutations as functionally impactful for the five proteins.

The next five sequence-based tools predict the disease phenotype (pathogenicity) of a single amino acid substitution. PON-P2, SNPs&GO, PhD-SNP, MutPred2 and PMut identify mutations as pathogenic or neutral. For PARK7, PARK2, PESN1, HRAS and RUNX1, the predicted pathogenic mutations by PON-P2 were 44.44%, 28.95%, 74.86%, 81.25% and 68.75%, respectively (**S1–S5 Figs**). Through PON-P2 identifier submission, we have obtained an average of 59.65% pathogenic mutations for the proteins. MutPred2 identifies the pathogenic mutations and tells the altering impact of a particular mutation on the protein structure. MutPred2 gives

**Table 3. Percentage of deleterious/ pathogenic/ destabilizing single point mutations predicted by all the fifteen tools for PARK7, PARK2, PESN1, HRAS and RUNX1.**

|  |  | Tools | PARK7 | PARK2 | PSN1 | HRAS | RUNX1 | Average |
|---|---|---|---|---|---|---|---|---|
| **Sequence-based** | Deleterious/Damaging | **PolyPhen2** | 66.67% | 92.11% | 96.09% | 78.13% | 100% | 86.60% |
|  |  | **PROVEAN** | 72.22% | 71.05% | 88.83% | 100% | 100% | 86.42% |
|  |  | **SIFT** | 66.67% | 78.95% | 90.50% | 90.63% | 100% | 85.35% |
|  |  | **FATHMM** | 50% | 63.16% | 68.16% | 53.13% | 87.5% | 64.39% |
|  |  | **Mutation Assessor** | 66.67% | 86.84% | 94.41% | 71.88% | 62.5% | 76.46% |
|  | Pathogenicity | **PON-P2** | 44.44% | 28.95% | 74.86% | 81.25% | 68.75% | 59.65% |
|  |  | **MutPred2** | 61.11% | 76.32% | 94.97% | 100% | 93.75% | 85.23% |
|  |  | **SNPs & GO** | 61.11% | 39.47% | 73.74% | 53.13% | 93.75% | 64.24% |
|  |  | **PhD-SNP** | 55.56% | 60.53% | 86.03% | 75% | 100% | 75.42% |
|  |  | **PMut** | 94.44% | 63.16% | 94.97% | 96.88% | 100% | 89.89% |
| **Structure-based** | Stability | **mCSM** | 88.89% | 94.74% | 87.71% | 93.75% | 100% | 93.02% |
|  |  | **SDM** | 83.33% | 81.58% | 63.69% | 62.50% | 62.5% | 70.72% |
|  |  | **MAESTROweb** | 88.89% | 63.16% | 79.33% | 84.38% | 81.25% | 79.40% |
|  |  | **CUPSAT** | 66.67% | 73.68% | 64.25% | 68.75% | 56.25% | 65.92% |
|  |  | **DynaMut2** | 83.33% | 89.47% | 82.68% | 84.38% | 87.5% | 85.47% |

structure altering mutations for PARK7, PARK2, PESN1, HRAS and RUNX1 estimated 61.11%, 76.32%, 94.97%, 100% and 93.75% pathogenic mutations, respectively. MutPred2 predicted an average of 85.23% pathogenic mutation for the five proteins. SNPs&GO uses gene ontology terms to predict disease-associated mutation. It also returns the PhD-SNP results along with the SNPs&GO results. Both tools predict the disease association through functional annotation. The average number of pathogenic mutations estimated by SNPs&GO was 64.24%, and the predicted average disease-associated mutations was 75.42% for PhD-SNP. PMut webserver predicted 94.44%, 63.16%, 94.97%, 96.88% and 100% pathogenic mutations for PARK7, PARK2, PESN1, HRAS and RUNX1, respectively.

After the sequence-based analysis, we performed the structure-based analysis of the mutations by five tools, namely mCSM, SDM, MAESTROweb, CUPSAT and DynaMut2. These tools provide Gibbs free energy change values (ΔΔG) for each protein structure; the change in free energy during the unfolding of a kinetically stable protein is described by this ΔΔG value. Sometimes the mutation in proteins differentiates the free energy landscape between the mutant and the WT protein. This variance in the free energy landscape is why the mutation affects the stability of a protein. Thermodynamically, the Gibbs free energy difference between folded (Gf) and unfolded (Gu) protein can be calculated as $\Delta G$ = Gu-Gf. The change of protein stability (ΔΔG) and free energy landscape between mutant (Gm) and WT (Gw) is calculated as $\Delta\Delta G$ = Gm-Gw. A negative ΔΔG value indicates stabilizing, and a positive ΔΔG shows destabilizing [38]. mCSM predicted an average of 93% mutations as destabilizing and 7% mutations as stabilizing for all five proteins. Predicted destabilizing mutations for PARK7, PARK2, PESN1, HRAS and RUNX1 by SDM were 83.33%, 81.58%, 63.69%, 62.50% and 62.5%, respectively, with an average of 70.72%. MAESTROweb estimated an average of 79.40% destabilizing mutations, whereas CUPSAT predicted an average of 65.92% destabilizing amino acid substitutions. The predicted destabilizing mutations for PARK7, PARK2, PESN1, HRAS and RUNX1 by DynaMut2 were 66.67%, 92.11%, 96.09%, 78.13% and 100%, respectively, with an average of 85.47%.

The sequence-based analysis using ten different tools revealed a comparative assessment of the tools. PolyPhen2, PROVEAN, FATHMM, SIFT and Mutation Assessor are the five sequence-based tools which categorize mutation into damaging/deleterious and tolerant categories. PolyPhen2, PROVEAN and SIFT showed almost equal prediction accuracy, whereas FATHMM showed a significant drop in average deleterious mutations with 64.39%. The other five sequence-based tools, PON-P2, SNPs&GO, PhD-SNP, MutPred2 and PMut, predicts the disease phenotype or pathogenicity of a single point mutation. PMut showed the highest average pathogenicity prediction with 89.9%. On the other hand, PON-P2 estimated the least average with 59.6%. MutPred2 showed higher accuracy than SNPs&GO and PhD-SNP. After the sequence-based tools, we compared the results of five structure-based tools mCSM, SDM, MAESTROweb, CUPSAT and DynaMut2. The structure-based tools predict the stabilizing or destabilizing mutations based on ΔΔG. mCSM predicted the highest number of mutations as destabilizing, whereas CUPSAT showed the least number of mutations as destabilizing (**Fig 2**).

## Conclusion

Single point amino-acid substitutions are associated with several human diseases, including cancer and neurodegenerative diseases, and are contemplated as one of the most recurrent genetic variants. Detailed analysis of the single point amino-acid substitution can help us understand the impact of mutation and the disease-causing mechanism. With a growing number of genetic variations, it is critical to predict the impact of a mutation through computational approaches in a fast and reliable manner. There are several computational methods

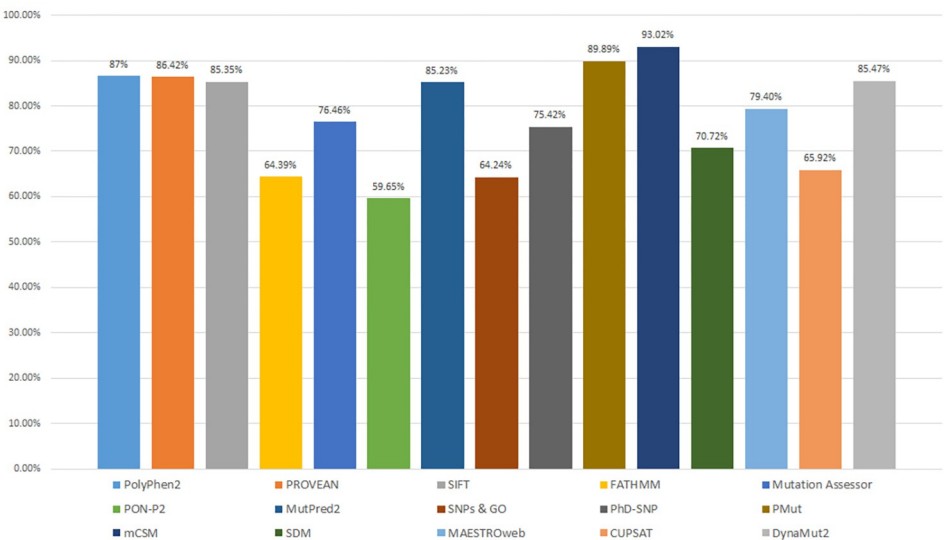

**Fig 2. Distribution of average deleterious/destabilizing single amino acid substitutions predicted by all fifteen tools for PARK7, PARK7, PESN1, HRAS and RUNX1.**

available to analyze the molecular consequences of single point mutations. We have performed a detailed analysis of several mutations through 15 different tools to determine the prediction accuracy based on previously available data. Out of the sequence-based tools that estimate deleterious/damaging mutation, we have found that PolyPhen2 and PROVEAN showed higher prediction accuracy. In sequence-based pathogenicity prediction, PMut showed the highest prediction accuracy. Out of the structure-based web tools, mCSM showed a higher number of mutations as destabilizing and showed higher prediction power than others. The results of this study may be used to designate the most suitable program for mutational analysis. An advanced platform then can be developed that can automatically select the program that is likely to give the most precise predictions.

## Supporting information

**S1 Data.**
(XLSX)

**S1 Fig. Distribution of deleterious/destabilizing mutations predicted by all 15 tools for PARK7.**
(DOCX)

**S2 Fig. Distribution of deleterious/destabilizing mutations predicted by all 15 tools for PARK2.**
(DOCX)

**S3 Fig. Distribution of deleterious/destabilizing mutations predicted by all 15 tools for PSN1.**
(DOCX)

**S4 Fig. Distribution of deleterious/destabilizing mutations predicted by all 15 tools for HRAS.**
(DOCX)

**S5 Fig. Distribution of deleterious/destabilizing mutations predicted by all 15 tools for RUNX1.**
(DOCX)

## Author Contributions

**Conceptualization:** Arunabh Choudhury, Taj Mohammad, Md. Imtaiyaz Hassan.

**Data curation:** Arunabh Choudhury, Bekhzod Abdullaev, Md. Imtaiyaz Hassan.

**Formal analysis:** Arunabh Choudhury, Bekhzod Abdullaev.

**Funding acquisition:** Farah Anjum.

**Investigation:** Taj Mohammad, Farah Anjum, Indrakant K. Singh, Dharmendra Kumar Yadav.

**Methodology:** Taj Mohammad, Indrakant K. Singh, Dharmendra Kumar Yadav, Md. Imtaiyaz Hassan.

**Project administration:** Taj Mohammad, Farah Anjum, Mohd Adnan.

**Resources:** Farah Anjum, Alaa Shafie, Bekhzod Abdullaev, Mohd Adnan, Dharmendra Kumar Yadav.

**Software:** Arunabh Choudhury, Farah Anjum, Alaa Shafie, Visweswara Rao Pasupuleti, Dharmendra Kumar Yadav.

**Supervision:** Alaa Shafie, Visweswara Rao Pasupuleti.

**Validation:** Arunabh Choudhury, Taj Mohammad, Alaa Shafie, Bekhzod Abdullaev, Mohd Adnan, Dharmendra Kumar Yadav, Md. Imtaiyaz Hassan.

**Visualization:** Taj Mohammad, Alaa Shafie, Indrakant K. Singh, Bekhzod Abdullaev.

**Writing – original draft:** Arunabh Choudhury, Taj Mohammad, Visweswara Rao Pasupuleti, Md. Imtaiyaz Hassan.

**Writing – review & editing:** Indrakant K. Singh, Visweswara Rao Pasupuleti, Mohd Adnan, Md. Imtaiyaz Hassan.

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
