## [Decision Letter · Decision Letter 0]

26 Dec 2021

PONE-D-21-35921Comparative analysis of web-based programs for single amino acid substitutions in proteinsPLOS ONE

Dear Dr. Hassan,

Thank you for submitting your manuscript to PLOS ONE. After careful consideration, we feel that it has merit but does not fully meet PLOS ONE’s publication criteria as it currently stands. Therefore, we invite you to submit a revised version of the manuscript that addresses the points raised during the review process.

The manuscript has been reviewed by three independent reviewers. They find merit in this manuscript but have highlighted several areas in which improvement and corrections are necessary. These areas include the organization of the text as well as technical details, and must be addressed for the manuscript to be considered for publication. Few language issues also need to be resolved by the authors.

We look forward to receiving your revised manuscript.

Kind regards,

Timir Tripathi, Ph.D.

Academic Editor

PLOS ONE

Journal Requirements:

3. We note you have included a table to which you do not refer in the text of your manuscript. Please ensure that you refer to Table 3 in your text; if accepted, production will need this reference to link the reader to the Table.

Reviewers' comments:

Reviewer's Responses to Questions

**Comments to the Author**

1. Is the manuscript technically sound, and do the data support the conclusions?

Reviewer #1: Yes

Reviewer #2: Yes

Reviewer #3: Yes

2. Has the statistical analysis been performed appropriately and rigorously? 

Reviewer #1: Yes

Reviewer #2: N/A

Reviewer #3: Yes

3. Have the authors made all data underlying the findings in their manuscript fully available?

Reviewer #1: Yes

Reviewer #2: Yes

Reviewer #3: Yes

4. Is the manuscript presented in an intelligible fashion and written in standard English?

Reviewer #1: Yes

Reviewer #2: Yes

Reviewer #3: Yes

5. Review Comments to the Author

Reviewer #1: 1. Introduction should be comprehensive. A discussion on the significance of these tools should be addressed.

2. The rationale to do such kind of review and analytical comparison must be described.

3. The calculations were performed on 15 different predictors. Are these tools based on the same prediction methods? Discuss this in the manuscript.

4. The authors used five different proteins in the study. What would be the results of calculations upon only one protein?

5. Were the average values estimated for all the parameters of each tool?

6. The explanation of the observed differences in the prediction power of each tool should be elaborated in light of their algorithms and resultant decision.

7. Authors double-check the manuscript for abbreviations used.

8. Language editing is required to improve the quality of the manuscript. The author should recheck this manuscript carefully and remove typos and grammatical errors.

9. All references should be thoroughly checked, and especially Author must confirm only relevant publications should be cited.

Reviewer #2: This study compares various available bioinformatics tools for predicting the impact of mutations in human proteins on their structure and function. The scope of the study is wide, with over a dozen computational web-based tools to evaluate their prediction power. The current form of the manuscript, unfortunately, ignores the following points which need to be resolved during revision.

· Despite interesting findings this paper lacks sound rationale and experimental support. The drawn conclusion should be focused and crisp.

· What is the basis of selecting five different proteins in this study? Is there any evidence that SNPs under consideration in this study are associated with disease? Discuss the rationale for this.

· Is there any clinical evidence showing that the destabilizing/deleterious nsSNPs are associated with protein dysfunction?

· How was the comparison of different predictors batched? A little detail of each predictor must be addressed.

· Method section may be shortened.

· Discussion should be improved in light of the author's findings and previous literature.

· The authors have not explained any relation between the prediction strategies of different predictors. This should be highlighted in the text.

· It is necessary to write the reasons why different five proteins were used as the benchmark.

· In conclusion, the authors write that “Out of the structure-based web tools, mCSM showed a higher number of mutations as destabilizing”. What does it mean? This statement should be elaborated.

· The results section is redundant. Please revise, focusing on the specific outcomes and their importance. The strength of the author’s findings should be highlighted.

· A uniform presentation is required. The author should proofread the manuscript before final submission.

Reviewer #3: 1. Abstract: Can be concise and trim-down; repetitive meanings should be avoided.

2. The scientific problem is described well, however, there are a few language mistakes in the text. Therefore, language editing is required to improve the quality of the manuscript. Grammatical mistakes and typographical errors should be corrected.

3. Introduction: Some of the English terminology used is odd which needs to be updated during the revision.

4. Introduction, second paragraph, first sentence, needs citation.

5. The results section has some redundancy in many parts. Authors should update this, focusing on the specific outcomes and their significance.

6. It is not clear how the destabilizing parameters compared with the values estimated by different tools based on totally different approaches?

7. The terms deleterious/pathogenic/destabilizing should be described clearly, and their correlations with the prediction should be discussed.

8. Table 1, a column mentioning the reference for the corresponding tool, should be added.

9. Discussion should be focused. The author should adhere to the results obtained from experiments.

10. Conclusion should be crisp and focused. Outcome must be highlighted in the conclusion section. Conclusions should provide be more details and further highlight the work and its potential importance/application

6. PLOS authors have the option to publish the peer review history of their article (what does this mean?). If published, this will include your full peer review and any attached files.

Reviewer #1: **Yes: **Tooba Naz Shamsi

Reviewer #2: No

Reviewer #3: **Yes: **Dr. Ethayathulla Abdul samath

---

## [Author Response · Author response to Decision Letter 0]

27 Mar 2022

Journal: PLOS ONE

Manuscript Title: Comparative analysis of web-based programs for single amino acid substitutions in proteins

Manuscript ID: PONE-D-21-35921

Reviewer #1:

1. Introduction should be comprehensive. A discussion on the significance of these tools should be addressed.

Response: Thank you for your valuable suggestion. Now we have discussed the introduction more compressively comprising the significance of each tool.

2. The rationale to do such kind of review and analytical comparison must be described.

Response: The rationale of the study has been described now.

3. The calculations were performed on 15 different predictors. Are these tools based on the same prediction methods? Discuss this in the manuscript.

Response: Different tools are not based on the same method. We have discussed this in the revised manuscript.

4. The authors used five different proteins in the study. What would be the results of calculations upon only one protein? 

Response: Five different proteins were used for high accuracy, involving the divorce datasets in the calculation. It minimizes the false prediction where only one protein can have less coherent outcomes.

5. Were the average values estimated for all the parameters of each tool?

Response: No, the prediction was considered as yes or no for destabilizing mutation.

6. The explanation of the observed differences in the prediction power of each tool should be elaborated in light of their algorithms and resultant decision. 

Response: Thank you for your valuable suggestion. Now we have discussed the prediction power of all the tools and their algorithms more compressively.

7. Authors double-check the manuscript for abbreviations used.

Response: The manuscript has been thoroughly checked for abbreviations and updated during this revision. Thanks! 

8. Language editing is required to improve the quality of the manuscript. The author should recheck this manuscript carefully and remove typos and grammatical errors.

Response: The manuscript has now been thoroughly checked and corrected for typos and language errors.

9. All references should be thoroughly checked, and especially Author must confirm only relevant publications should be cited.

Response: The reference section has been updated now.

Reviewer #2:

This study compares various available bioinformatics tools for predicting the impact of mutations in human proteins on their structure and function. The scope of the study is wide, with over a dozen computational web-based tools to evaluate their prediction power. The current form of the manuscript, unfortunately, ignores the following points which need to be resolved during revision.

• Despite interesting findings this paper lacks sound rationale and experimental support. The drawn conclusion should be focused and crisp.

Response: Thank you for your valuable suggestion. Now we have highlighted the rationale and discussion part of the revised manuscript.

• What is the basis of selecting five different proteins in this study? Is there any evidence that SNPs under consideration in this study are associated with disease? Discuss the rationale for this.

Response: Multiple proteins were used to identify diseased mutations since studying only one protein can provide some false positives. To avoid any false prediction, multiple datasets were used, warranting more accuracy of the outcomes. Yes, there are several evidences SNPs under consideration are associated with disease progression. The text has been updated in the revised manuscript.

• Is there any clinical evidence showing that the destabilizing/deleterious nsSNPs are associated with protein dysfunction?

Response: There are several clinical findings that the destabilizing/deleterious nsSNPs of the selected proteins are associated with protein dysfunction resulting in disease progression. We have updated the discussion part of the revised manuscript.

• How was the comparison of different predictors batched? A little detail of each predictor must be addressed.

Response: The prediction was considered as an independent decision of each tool. Now we have described each predictor during this revision.

• Method section may be shortened.

Response: The method section has already been written briefly. Shortening the method section may cause intricacy for the readers.

• Discussion should be improved in light of the author's findings and previous literature. 

Response: Thank you for this valuable suggestion; the discussion section has now been updated.

• The authors have not explained any relation between the prediction strategies of different predictors. This should be highlighted in the text.

Response: We have used different tools for detecting the pathogenicity of the variations. The relation between the prediction strategies of these predictors has been highlighted in the revised manuscript. 

• It is necessary to write the reasons why different five proteins were used as the benchmark.

Response: Five different proteins were used to avoid any false prediction; only one protein can provide some false positives. Now the reason for this has been updated in the revised manuscript.

• In conclusion, the authors write that “Out of the structure-based web tools, mCSM showed a higher number of mutations as destabilizing”. What does it mean? This statement should be elaborated.

Response: We have studied only destabilizing/diseased mutations to compare the predictive power of each tool. Here mCSM showed a higher number of mutations as destabilizing means having higher prediction power than others. We have discussed this in more detail during this revision.

• The results section is redundant. Please revise, focusing on the specific outcomes and their importance. The strength of the author’s findings should be highlighted.

Response: The result section has been updated in light of the reviewers comment.

• A uniform presentation is required. The author should proofread the manuscript before final submission.

Response: The manuscript has been thoroughly checked and updated during this revised submission.

Reviewer #3:

Comments to the Authors:

1. Abstract: Can be concise and trim-down; repetitive meanings should be avoided.

Response: Thank you for your suggestion. Now the abstract of the manuscript has been updated during this revision. 

2. The scientific problem is described well, however, there are a few language mistakes in the text. Therefore, language editing is required to improve the quality of the manuscript. Grammatical mistakes and typographical errors should be corrected.

Response: The manuscript has now been thoroughly checked and corrected for typos and language errors.

3. Introduction: Some of the English terminology used is odd which needs to be updated during the revision.

Response: Now, the introduction section has been checked for any odd terminology and updated in this revised submission.

4. Introduction, second paragraph, first sentence, needs citation.

Response: The citation has now been added to the respective section.

5. The results section has some redundancy in many parts. Authors should update this, focusing on the specific outcomes and their significance.

Response: The result section has been revised as per the reviewer's suggestion. We have updated this in a more comprehensive way now.

6. It is not clear how the destabilizing parameters were compared with the values estimated by different tools based on totally different approaches? 

Response: Each tool gives its score based on some calculations and predicts either a mutation in a protein is destabilizing or not. This has been discussed in the revised submission.

7. The terms deleterious/pathogenic/destabilizing should be described clearly, and their correlations with the prediction should be discussed. 

Response: Thank you for your suggestion. Now we have described these terms and discussed their correlation with the prediction in the revised submission.

8. Table 1, a column mentioning the reference for the corresponding tool, should be added.

Response: Added

9. Discussion should be focused. The author should adhere to the results obtained from experiments.

Response: The discussion part of the manuscript has been revised now.

10. Conclusion should be crisp and focused. Outcome must be highlighted in the conclusion section. Conclusions should provide be more details and further highlight the work and its potential importance/application.

Response: Thank you for your suggestion. Now we have updated the conclusion of the revised manuscript.

---

## [Decision Letter · Decision Letter 1]

4 Apr 2022

Comparative analysis of web-based programs for single amino acid substitutions in proteins

PONE-D-21-35921R1

Dear Dr. Hassan,

We’re pleased to inform you that your manuscript has been judged scientifically suitable for publication and will be formally accepted for publication once it meets all outstanding technical requirements.

Kind regards,

Timir Tripathi, Ph.D.

Academic Editor

PLOS ONE

Additional Editor Comments (optional):

Reviewers' comments:

Reviewer's Responses to Questions

**Comments to the Author**

1. If the authors have adequately addressed your comments raised in a previous round of review and you feel that this manuscript is now acceptable for publication, you may indicate that here to bypass the “Comments to the Author” section, enter your conflict of interest statement in the “Confidential to Editor” section, and submit your "Accept" recommendation.

Reviewer #1: All comments have been addressed

Reviewer #3: All comments have been addressed

2. Is the manuscript technically sound, and do the data support the conclusions?

Reviewer #1: Yes

Reviewer #3: Yes

3. Has the statistical analysis been performed appropriately and rigorously? 

Reviewer #1: Yes

Reviewer #3: Yes

4. Have the authors made all data underlying the findings in their manuscript fully available?

Reviewer #1: Yes

Reviewer #3: Yes

5. Is the manuscript presented in an intelligible fashion and written in standard English?

Reviewer #1: Yes

Reviewer #3: Yes

6. Review Comments to the Author

Reviewer #1: The authors have addressed all the suggestions and fixed them very well. The revised manuscript should be accepted as is.

Reviewer #3: The authors have addressed all the concerns and queries in the manuscript. Accordingly to my review it can be accepted in the current status

7. PLOS authors have the option to publish the peer review history of their article (what does this mean?). If published, this will include your full peer review and any attached files.

Reviewer #1: **Yes: **Tooba Naz Shamsi

Reviewer #3: **Yes: **Ethayathulla Abdul Abdul samath

---

## [Editor Report · Acceptance letter]

25 Apr 2022

PONE-D-21-35921R1 

Comparative analysis of web-based programs for single amino acid substitutions in proteins 

Dear Dr. Hassan:

I'm pleased to inform you that your manuscript has been deemed suitable for publication in PLOS ONE. Congratulations! Your manuscript is now with our production department. 

Kind regards, 

on behalf of

Dr. Timir Tripathi 

Academic Editor

PLOS ONE